



# The High-resolution Intermediate Complexity Atmospheric Research (HICAR v1.0) Model Enables Fast Dynamic Downscaling to the Hectometer Scale

Dylan Reynolds[1,2], Ethan Gutmann[3], Bert Kruyt[3,4], Michael Haugeneder[1], Tobias Jonas[1], Franziska Gerber[1,2], Michael Lehning[1,2], and Rebecca Mott[1]

[1]Institute for Snow and Avalanche Research SLF, Davos, Switzerland
[2]School of Architecture, Civil and Environmental Engineering, Ècole Polytechnique Fèdèrale de Lausanne, Lausanne, Switzerland
[3]Research Applications Laboratory, National Center for Atmospheric Research, Boulder, Colorado
[4]Subzero Research Laboratory, Montana State University, Bozeman MT, USA

**Correspondence:** Dylan Reynolds (dylan.reynolds@slf.ch)

**Abstract.**

High resolution (< 1km) atmospheric modeling is increasingly used to study precipitation distributions in complex terrain and cryosphere-atmospheric processes. While this approach has yielded insightful results, studies over annual time-scales or at the spatial extents of watersheds remain unrealistic due to the computational costs of running most atmospheric models.

In this paper we introduce a High-resolution variant of the Intermediate Complexity Atmospheric Research (ICAR) model, HICAR. We detail the model development that enabled HICAR simulations at the hectometer scale, including changes to the advection scheme and the wind solver. The latter uses near surface terrain parameters which allow HICAR to simulate complex topographic flow features. These model improvements clearly influence precipitation distributions at the ridge scale (50m), suggesting that HICAR can approximate processes dependent on particle-flow interactions such as preferential deposition. A 250 m HICAR simulation over most of the Swiss Alps also shows monthly precipitation patterns similar to two different gridded precipitation products which assimilate available observations. Benchmarking runs show that HICAR uses 118x fewer computational resources than the WRF atmospheric model. This gain in efficiency makes dynamic downscaling accessible to ecohydrological research, where downscaled data is often required at hectometer resolution for whole basins at seasonal time scales. These results motivate further development of HICAR, including refinement of parameterizations used in the wind solver, and coupling of the model with an intermediate complexity snow model.

## 1 Introduction

Atmospheric models have seen remarkable improvements over the past decades, spurred on by their importance to society. Their usage within science ranges from climate and weather predictions to downscaling atmospheric variables as input to further geophysical models. Specific applications have included generating forcing data over sparsely instrumented domains (Khadka et al., 2022), downscaling global climate model output to study regional impacts (Spinoni et al., 2018), and coupling





with land surface models to better simulate land-atmosphere feedbacks (Sharma et al., 2023). The concept intrinsic to all of these applications is one of scale. As model resolution increases, processes which were previously parameterized can be explicitly resolved, and the representation of the underlying terrain improves, allowing for more accurate dynamics (Wyngaard, 2004; Chow et al., 2019; Prein et al., 2013).

High-resolution (< 1km) simulations of winter storms in complex terrain have been used to augment our process-level understanding of particle-flow interactions such as preferential deposition (Lehning et al., 2008; Gerber et al., 2018; Vionnet et al., 2017; Mott et al., 2010). Some of these simulations aimed at very high resolutions of 25 m and below and thus used stationary wind fields (Raderschall et al., 2008) or a decomposition of wind field into a limited number of dominating (stationary) patterns to enable simulations for the length of a storm (Mott et al., 2010) to a full season (Groot Zwaaftink et al., 2013). Coupled
glacier-atmosphere models have been developed and run at a range of spatial scales, demonstrating an ability to better simulate surface-atmosphere energy exchanges over glaciers (Collier et al., 2013; Goger et al., 2022). And, coupled snow-atmosphere models have been developed which explicitly resolve snow-atmosphere interactions (Vionnet et al., 2014; Sharma et al., 2023). These studies have all demonstrated the ability of high-resolution atmospheric modeling to improve estimates of precipitation, wind speeds, and surface-atmosphere interactions. However, all of them have focused on limited spatial and temporal extents
due to the huge computational demand required of running modern atmospheric models at the hectometer resolution. In one study performing 50m simulations of winter precipitation using the WRF model, nearly 34,000 core hours were required to perform 1 day of simulation over a <100km$^2$ domain (Kruyt et al., 2022). Any practical application of high-resolution atmospheric modeling to questions concerning future climate scenarios or downscaling for land surface models is currently limited by the computational demand of atmospheric models.

This issue is no news to the community, and idealized atmospheric models of orographic precipitation and mountain waves have been developed and employed in the past (Smith, 1979; Smith and Barstad, 2004). Recently, the Intermediate Complexity Atmospheric Research (ICAR) model was introduced in Gutmann et al. (2016) (hereafter G16) to provide an alternative to highly idealized models and modern non-hydrostatic, compressible atmospheric models. In their 2016 paper, Gutmann et al., demonstrated excellent agreement between ICAR and WRF when simulating mountain waves and orographic precipitation over
idealized terrain. Further demonstration over real, complex terrain at a 4km resolution gave good agreement on precipitation between the two models during the winter months. Most importantly, the ICAR simulations used 143x fewer computational resources than the WRF model. The ability of ICAR to simulate orographic precipitation at the kilometer-scale has been replicated in other studies (Horak et al., 2019). ICAR has since occupied a niche in modeling studies where downscaling of long time series would otherwise be limited by computational resources. These results motivate the design philosophy behind
ICAR that dramatic reductions in computational time may justify modest reductions in model accuracy for certain applications.

Such an approach is perfectly suited for high-resolution atmospheric modeling, where computational demands severely limit the experimental design of studies. However, the dynamics and physics of the base ICAR model, namely linear mountain wave theory and first-order upwind advection, are not suitable when modeling at the hectometer scale. Here we introduce a High-resolution variant of the ICAR model, HICAR, which adapts the ICAR model to be suitable at resolutions below the
kilometer scale. In the second section of the paper, key parts of HICAR's model development are detailed, with a focus on the





model's wind solver, advection scheme, and input/output (I/O) operations. In the third section, information is given about other atmospheric models and gridded datasets used in this study, as well as details about model simulation setups. These models and datasets are then compared in section four, where various demonstrations of the HICAR model provide a limited validation and are used to discuss the model performance. Lastly, a synthesis of the paper and a concluding discussion about the utility of

the HICAR model is presented in section five.

## 2   Model Development

In the original ICAR model, the 3-D wind field can either be generated through 3-D interpolation between the coarse resolution forcing data and the high-resolution grid, or it can be further modified using linear mountain wave theory (Smith, 1979). This modification alone simulates the disturbance of the meso-scale flow field caused by mountain ranges, namely the generation

of mountain waves depending on the atmospheric stability. These effects are the dominant influence of the terrain on the meso-scale flow from scales of 10s of kilometers down to the kilometer scale, which is the scale range which ICAR was originally developed for. Increasingly, output from kilometer-scale compressible, non-hydrostatic atmospheric models run by regional weather forecasting offices are available (Benjamin et al., 2016; Seifert et al., 2008; Seity et al., 2011). These models are expected to capture the dynamics approximated by linear mountain wave theory. When using these models as forcing

data for high-resolution simulations with ICAR, it would thus be redundant to run with the linear theory solution. Left with only an interpolated kilometer-scale wind field for a 3-D wind field, we found it necessary to implement a new wind solver capable of capturing dynamics induced by the underlying high-resolution terrain. These flow features should be necessary to simulate particle-flow interactions which lead to heterogeneous snowfall patterns. In addition to changes to the wind field, it was also necessary to modify the advection scheme of ICAR and the input/output (I/O) routines. ICAR only offers the first

order upwind advection scheme, which has been shown to be highly diffusive, especially in complex terrain (Schär et al., 2002). When simulating precipitation events, it is important that heterogeneities in moisture and temperature are maintained and do not become too smooth. Finally, as model resolution and speed increased, it became paramount to be able to efficiently read and write large volumes of data without significantly affecting run time. The following two subsections focus on new options for the wind solver in HICAR, while the last two focus on changes affecting the advection scheme and model input/output

(I/O)(Figure 1).

### 2.1   Direct Adjustment of Wind Field

Taking a cue from existing statistical models of surface winds in complex terrain (Winstral and Marks, 2002; Winstral et al., 2017; Liston and Elder, 2006; Dujardin and Lehning, 2022), we first develop corrections to the interpolated wind field near the surface based on the underlying terrain. This is done through terrain descriptors calculated at model initialization and then

applied to the wind field at runtime. Terrain descriptors represent some qualitative information about the terrain quantitatively, such as if a particular location is sheltered from a particular wind direction. Parameterizations can then be developed using these





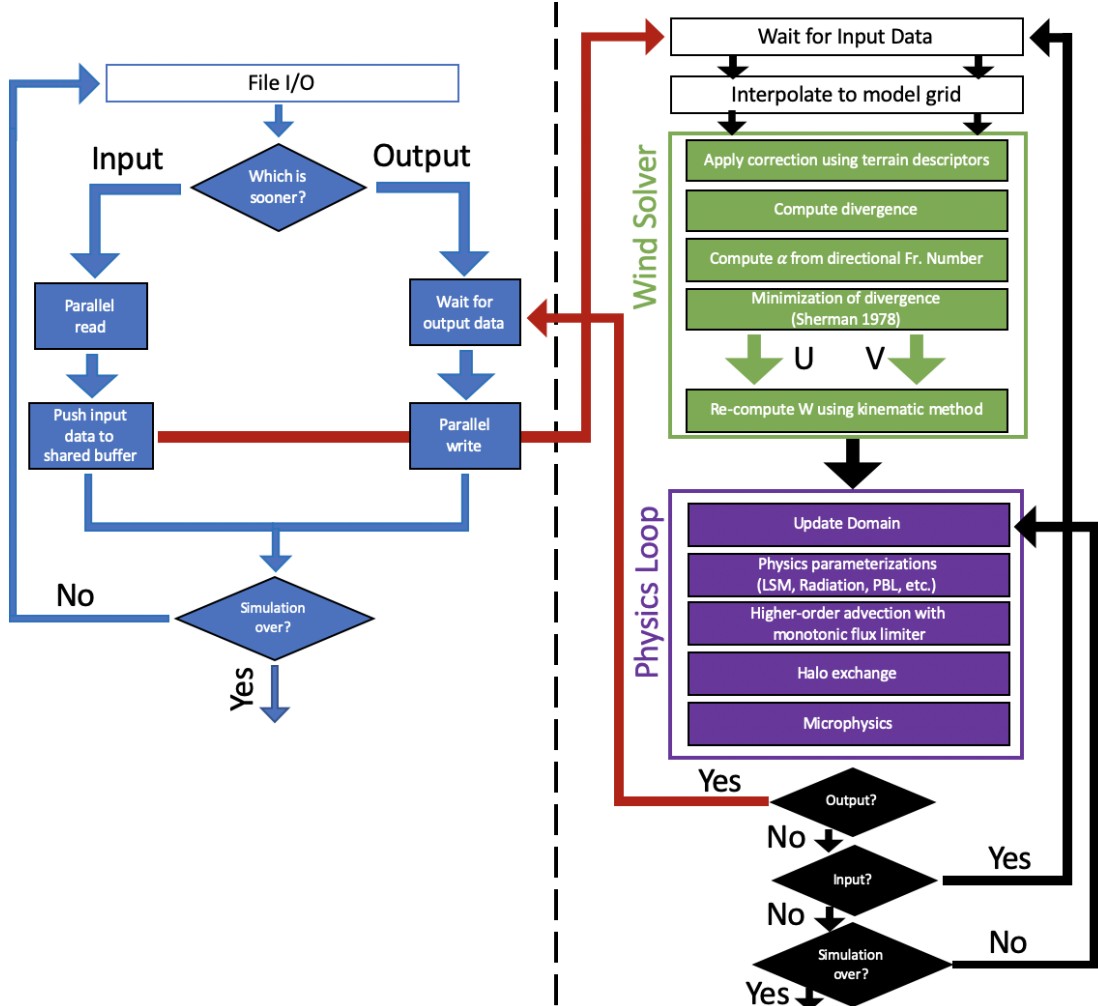

**Figure 1.** Schematic of major changes to HICAR's runtime loop compared to figure 1 of G16. The left side of the figure features the I/O loop handeled by I/O processes, while the right side features the runtime loop of HICAR, with a focus on the steps discussed in sections 2.2 and 2.3. Blue colors correspond to I/O processes, green to steps of the wind solver, purple to steps of the physics integration loop, and red to communication between I/O and compute processes. Within the wind solver and physics loop, downward arrows are implied between the steps where not indicated.

values, enabling non-local interactions between the topography and winds to be accounted for in a computationally efficient manner.

### 2.1.1 Terrain Descriptors

**Topographic Position Index (TPI)**





When downscaling winds from coarse to high resolutions, the representation of the model terrain can vary drastically. What appears as a small depression in the terrain at a 1km resolution may actually be a steep valley when viewed at a 100m resolution. To find areas in the high-resolution domain where large differences with the coarse Digital Elevation Model (DEM) may affect wind fields, we use the Topographic Position Index (TPI, Jenness 2006, Weiss 2001). TPI is calculated as the difference in elevation between a given terrain element, and the average terrain height within a given radius around that terrain element:

$$TPI = z_{\mathrm{hi}} - \bar{z}_{\mathrm{radius}} \tag{1}$$

Where $z_{\mathrm{hi}}$ is the high-resolution elevation and $\bar{z}_{\mathrm{radius}}$ is the mean elevation of the high-resolution grid within a given radius around $z_{\mathrm{hi}}$. We set the search radius to be 4 km. The chosen search radius will depend upon the resolutions of the model and the forcing data being used. In general, larger search radii lead to wider bands of positive and negative TPI, while smaller radii select just the valley bottoms and tops of peaks, resulting in a more heterogeneous distribution of TPI (Weiss 2001). TPI has previously been used as a variable in other wind downscaling schemes (Winstral et al., 2017), serving to highlight areas where winds are expected to be higher, such as an exposed ridge. TPI was chosen as a terrain descriptor instead of locally differencing the model and forcing DEMs because it gives a description of exposure, which is a non-local concept. For example, a hill in a valley may have the same elevation on the high-resolution grid as on the smoother, coarse-resolution forcing grid, and the terrain difference would be 0. However, if this hill is in a valley, it is still relatively lower than the surrounding terrain, and this would result in a negative TPI.

**3D Sx**

The Sx parameter was first introduced by Marks et al. (2002), quantifying the maximum slope from a surface grid cell to a terrain element in the upwind direction. The Sx parameter was thus interpreted as a proxy for how sheltered a surface grid cell was from incoming winds, as the upwind terrain element was expected to disrupt the flow. Sx has since been used in many parameterizations of surface wind (Marks et al., 2002; Winstral et al., 2013; Grünewald et al., 2013). Importantly, the Sx parameter gives directional information about terrain-wind interactions, which supplements the omni-directional TPI. Here we extend the original concept of Marks et al. (2002) into three dimensions, calculating Sx not just for the surface grid cells, but for all model grid cells in the vertical dimension. The motivation behind this is that the sheltering effects provided by an upwind terrain element will be felt above the surface as well as on the ground. The procedure for calculating 3D Sx is similar to that for 2D Sx: it is the maximum upwind slope between a grid cell (this time allowed to be above the surface) and the largest upwind terrain element. We add an important caveat that the largest upwind terrain element must also have a positive TPI value. This is done under the assumption that flow separation is more likely to occur for exposed terrain elements (positive TPI). The following equation:

$$Sx_{\mathrm{A,d_{max}}}(x,y,z) = max\left( tan^{-1}\left( \frac{DEM(x_{\mathrm{v}}, y_{\mathrm{v}}) - Z(x,y,z)}{\sqrt{(x_{\mathrm{v}} - x)^2 + (y_{\mathrm{v}} - y)^2}} \right) \right) \tag{2}$$





gives the Sx value for a given azimuth angle $A$, calculated at a specific point $(x,y,z)$, using a search radius of $d_{max}$. $DEM$ is the high-resolution DEM (2D) and $Z$ is the grid cell height on the mass grid (3D). $(x_v,y_v)$ give the location of the terrain element for which Sx is being calculated against. $d_{max}$ is a namelist variable which the user can define. A qualitative illustration of the 3D Sx parameter is given in Figure 2.

## 2.1.2 Application of Terrain Descriptors

The two terrain descriptors, TPI and Sx, seek to highlight areas of the domain where direct adjustment to the interpolated wind field are necessary. TPI indicates relative differences between the high-resolution terrain and a low-resolution representation, which is to say areas where the interpolated, high-resolution wind field are experiencing terrain features which the forcing terrain's lower resolution DEM may not resolve. Because TPI is non-directional, we only consider adjustments to the wind speed, and consider to increase wind speeds at areas of positive TPI (HICAR terrain higher than forcing terrain) and decrease them at areas of negative TPI. Testing showed that the wind solver discussed in section 2.2 adequately increases wind speeds over areas of positive TPI without a direct TPI-based adjustment, so only adjustments in areas of negative TPI are performed. This can be explained conceptually as reducing wind speeds in valleys deeper, and thus more removed from mesoscale wind speeds, than the forcing terrain suggests. This correction is only considered within the first 200m above the surface and is gradually decreased up to this height. This height limit was chosen empirically after testing multiple decay heights. Corrections based on TPI can thus be formulated as:

$$TPI_{cor} = \frac{TPI}{TPI_{max}} \frac{z_{top} - z}{z_{top}}, \qquad TPI < 0 \tag{3}$$

where $TPI$ is the surface TPI computed at each grid cell and $z$ is the height of the grid cell in question. $TPI_{max}$ is a scaling factor controlling the correction, and was set to 200 in our simulations. $z_{top}$ controls the height at which the correction goes to 0, in this case 200m.

Corrections based on the Sx parameter are considered for all grid cells with a negative Sx value. For these cells, a threshold Sx angle, $Sx_{thresh}$, is calculated at the surface:

$$N = \sqrt{\frac{g}{\theta} \frac{d\theta}{dz}} \tag{4}$$

$$Ri = \frac{N^2}{\left(\frac{du}{dz}\right)^2 + \left(\frac{dv}{dz}\right)^2} \tag{5}$$

$$Sx_{thresh} = 180° min\left(max\left(0, Ri\right), 0.25\right) \tag{6}$$

where $N$ is the Brunt-Väisälä frequency, $\theta$ is potential temperature, and $Ri$ is the Richardson Number. All vertical gradients are calculated over the first 100m above the surface. This is following the methodology of Menke et al. (2019) where the Richardson number used to classify stable and unstable conditions for leeside re-circulation was calculated over the first 100m above the surface. EQ #6 says that for $Ri$ values greater than 0.25 [Stable], no sheltering effects occur, and for negative $Ri$





values [Unstable], the threshold Sx angle is 0°. Although $Sx_{thresh}$ is only calculated at the surface, it is used throughout the column to apply the following corrections in 3D. This threshold angle is then used to calculate an Sx correction factor

$$Sx_{corr} = \frac{Sx - Sx_{thresh}}{\phi_{\text{def}}} \tag{7}$$

Where $Sx$ is the Sx angle for the given grid cell, $Sx_{thresh}$ is the threshold angle calculated for that column, and $\phi_{\text{def}}$, a scaling factor, is set to 30°. $Sx_{\text{corr}}$ is then applied to the $U$ and $V$ wind vectors by divvying up the correction according to the slope of the underlying topography. This is shown conceptually in Figure 2, and follows the equation:

$$SLOPE = \sqrt{\left(\frac{dz}{dx}\right)^2 + \left(\frac{dz}{dy}\right)^2} \tag{8}$$

$$Sx_{u,cor} = -\frac{dz}{dx}\frac{Sx_{cor}}{SLOPE^2}\left(\frac{dz}{dx}U_m + \frac{dz}{dy}V_m\right) \tag{9}$$

$$Sx_{v,cor} = -\frac{dz}{dy}\frac{Sx_{cor}}{SLOPE^2}\left(\frac{dz}{dx}U_m + \frac{dz}{dy}V_m\right) \tag{10}$$

Where $U_{\text{m}}$ and $V_{\text{m}}$ are the $U$ and $V$ velocities staggered to the mass-grid, and $SLOPE$ is the terrain slope. Vertical gradients shown here are calculated over the grid cell. The net effect is to apply both a correction to the wind speed, and to rotate the wind vector about the slope-tangent. Finally, the two correction factors for TPI and Sx are applied as such:

$$U = U - Sx_{u,corr} \tag{11}$$

$$V = V - Sx_{v,corr} \tag{12}$$

$$U = U(1 + TPI_{cor}) \tag{13}$$

$$V = V(1 + TPI_{cor}) \tag{14}$$

We note that parameter values and correction formulations used in this section are somewhat arbitrary. The logic behind the corrections is explained above, and the exact values were reached through a sparse sampling of the parameter space. The goal of the current study is to demonstrate the potential of combining a pre-conditioning step, described in the current section, with the diagnostic wind solver described in the following section. The effects of this currently under-constrained approach to correcting the wind field is discussed further in Section 4.1, and these corrections will be further refined in a future study by using observations of the 3D wind field in complex terrain.

## 2.2 Mass-Conserving Wind Solver

After adjusting the wind field according to terrain descriptors, or after ingesting any arbitrary wind field from forcing data, the resultant wind field is not guaranteed to be divergence-free. Because ICAR is an incompressible atmospheric model, this would

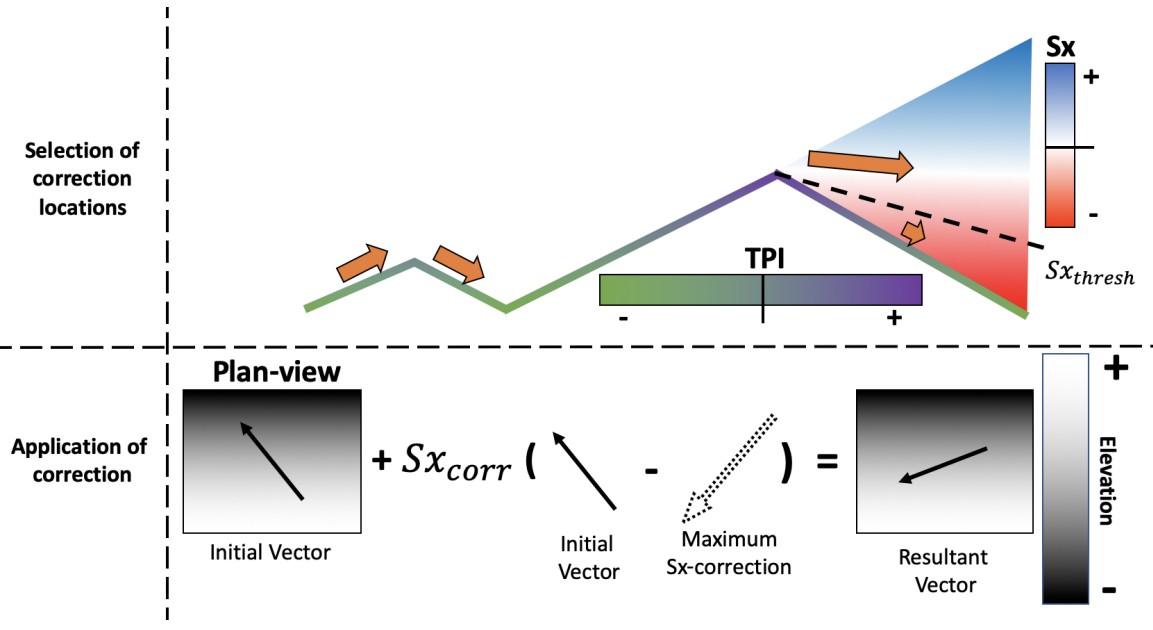

**Figure 2.** A conceptual outline of the Sx sheltering process. Areas where a correction should be applied are first selected, as indicated in the upper row. Only terrain elements with a positive TPI value are considered to be potential sheltering terrain elements. The smaller hill on the left has no positive TPI values along its slopes, so it does not produce an area of reduced wind speeds in the lee. The hill on the right does have a positive TPI value at its peak, so it is considered for sheltering. The Sx values in the leeside of the peak are examined and compared to the threshold Sx value, $Sx_{thresh}$, calculated in Eq. #6. Grid cells with Sx angles larger than this threshold angle experience a correction to their U and V wind speeds, as detailed in the second row of the figure. We consider that the maximum deflection of the leeside vector would be a rotation about the elevation gradient of the grid cell. This maximum correction is then applied to the initial vector with a correction factor, $Sx_{corr}$, as calculated in Eq. #7. The resultant vector is thus a mixture between the initial vector and the maximum possible correction.

175 mean a violation of mass-conservation. Thus, some further correction to the 3D wind field must be applied to ensure mass-conservation. In the original ICAR model, this is ensured by calculating the divergence for each model layer and prescribing the grid-relative vertical velocity at the top of each layer such that divergence is eliminated. This is sometimes referred to as the "kinematic method" of balancing the winds (O'brien, 1970; Homicz, 2002). Unfortunately, this method is known to produce excessive vertical motion even for modest amounts of residual divergence (Goodin et al., 1980). Figure 3 shows the strong

180 vertical winds which are often observed in high-resolution simulations using the ICAR model with the kinematic method for balancing the 3D wind field. The strong vertical winds observed in the ICAR simulations are due to a) large grid distortions in complex terrain at high resolutions, b) the use of high-resolution forcing data from a compressible atmospheric model, and c) the kinematic solution for vertical wind itself (EQ #9 in G16). As the horizontal resolution is reduced, the magnitude and variations of the vertical motions are reduced. As a result, simulations with the ICAR model at coarser resolutions exhibit less

185 strong vertical motion than shown here. However, such simulations still exhibit increasing vertical motion as a function of height due to the use of the kinematic solution for vertical velocity (O'brien, 1970). This results in excessively strong vertical



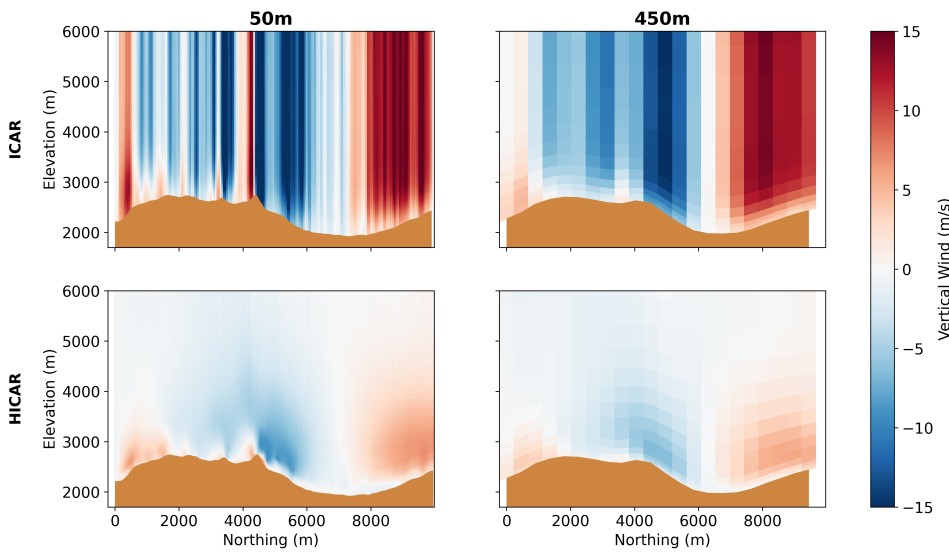

**Figure 3.** Comparison of vertical motion between ICAR and HICAR at 50m and 450m resolutions for an arbitrary simulation time step. ICAR is shown in the first row, HICAR in the second.

motion at the model top, and explains the sensitivity of ICAR to the height of the model top and choice of upper boundary condition reported in Horak et al. (2019) and Horak et al. (2021).

This issue alone motivates the implementation of a new approach to balancing the 3D wind field. When using the empirical adjustment of the 3D wind field described above, even more divergence is introduced to the wind field, resulting in entirely nonphysical vertical velocities. Clearly another technique for calculating vertical velocity is required for high-resolution applications.

HICAR employs a method for calculating a mass-conserving wind field which is based on a variational calculus technique. This technique has been developed over prior decades of wind modeling and pollutant transport (Sasaki, 1958; Sherman, 1978; Ross and Fox, 1991), and has been adapted into a variety of wind models (Moussiopoulos et al., 1988; Forthofer et al., 2014). Wind Tunnel experiments and field observations have routinely demonstrated this techniques ability to simulate speed up and deflection of flow around obstacles (Ross and Fox, 1991; Forthofer et al., 2014; Wagenbrenner et al., 2016). The method works by solving an optimization problem where two functions are reduced: the divergence of the wind field and the total deviations of the solution wind field from the initial wind field.

$$Div = \frac{d\rho u}{dx} + \frac{d\rho v}{dy} + \frac{d\rho \dot{w}}{d\dot{z}} \qquad (15)$$

$$Diff = (u_i - u)^2 + (v_i - v)^2 + \alpha(w_i - w)^2 \qquad (16)$$





Where $u$ and $v$ refer to the east- and north-ward wind speeds, $w$ refers to the vertical wind speed, and $\dot{w}$ refers to the contravariant, grid-relative wind speed. All of the $x_i$ variables indicate initial values. The distinction between $w$ and $\dot{w}$ is necessary when the optimization is performed on a grid with a vertical coordinate transformation such as sigma or SLEVE

coordinates (Gal-Chen and Somerville, 1975; Schär et al., 2002) and is further detailed in Ross et al. (1988). An excellent overview of the maths used to solve this optimization problem and a discussion of various considerations is given in Homicz (2002) and a general review is provided by Ratto et al. (1994). Because an initial guess is required for $w_i$, HICAR allows the user to specify vertical motion as an input variable. Otherwise, $w_i$ is taken to be 0, such that vertical motion is minimized. In the above equations, the variable $\alpha$ is used to control the relative weighting of changes to horizontal or vertical motion. This allows

the solution to account for effects of atmospheric stability if one makes $\alpha$ a function of atmospheric stability. For example, larger values of $\alpha$ increase the weighting of changes to $w$ from its initial value relative to changes of $u$ and $v$ from their initial values. This means that a better solution to the minimization would be found by preferring changes to $u$ and $v$ over $w$ when eliminating divergence. The result of this is more deflection around terrain and less vertical motion, which one would expect during stable atmospheric conditions. A demonstration of the effects of different values of $\alpha$ is given in Figure 4, showing

the wind field generated by the maximum (1.0) and minimum (0.1) values that $\alpha$ is allowed to take. For the stable condition ($\alpha = 1.0$) we see surface wind speeds approaching 10 m s-1 over the ridge crest and blocking of flow upwind of the ridge. Correspondingly, vertical motion is around +/- 2 m s-1 over the ridge. For the unstable case ($\alpha = 0.1$), there is comparatively little deflection of the flow field upwind of the ridge, and little speed up over the ridge crest. Vertical motion is significantly enhanced in the unstable case versus the stable case. As such, $\alpha$ can be used to select different solutions to the optimization

problem depending on atmospheric stability.

In our implementation, the $\alpha$ variable is calculated at each input time step and for each grid cell according to the atmospheric stability at that location according to:

$$\alpha = \sqrt{1 - 0.5 \frac{\sqrt{1 + 4Fr^4} - 1}{Fr^4}} \tag{17}$$

$$Fr = \frac{WS}{L * N} \tag{18}$$

Where Fr is the Froude number, WS is the wind speed, L is the scale length, and N is the Brunt-Väisälä frequency (BVF). Equation #17 comes from Moussiopoulos et al. (1988) and is straight forward, but the calculation of the Froude number deserves further discussion. In order to calculate $\alpha$ in 3D, the Froude number must also be calculated in 3D. To do this, WS, L, and N are calculated for each grid cell. The scale length, L, is the height difference between the grid cell height and the largest downwind terrain element, plus some constant to ensure a minimum value for L. L is calculated for each grid cell and each wind

direction at initialization so that it can be easily looked up at run time. Some search radius must be imposed when calculating L, which we set to 4km. Brunt-Väisälä frequency is then calculated by considering the column of air above the grid cell for which it is calculated. If there is a downwind obstacle, the column of air extends from the current grid cell height up to the altitude of the downwind obstacle. If there is no obstacle, BVF is calculated using a difference over the current grid cell. The effect





of these considerations is a Froude number which describes the ease of lifting a parcel of air over a given downwind obstacle. This approach of using a spatial-temporally varying $\alpha$ differs from prior implementations of Sherman 1978's technique, where either $\alpha$ was set to be 1.0 (Forthofer et al., 2014) or where $\alpha$ varied in time but not in space (Moussiopoulos et al., 1988). Thus our approach can handle complex situations where flow blocking varies as a function of height, such that flow may be blocked at the foot of a mountain but rise over the obstacle at higher altitudes. The computational demands of this technique are relatively small in comparison to other components of HICAR (advection, microphysics), since most of its calculations are performed once at initialization, and the solution of equations # 15 and 16 are only performed when ingesting new input data instead of at every physics time step.

### 2.3 Advection and Physics Parameterizations

The original ICAR model offers a first-order upwind advection scheme. Although this scheme is highly diffusive (Schär et al., 2002), it has the advantage of low computational demand, making it suitable for ICAR's original development purposes and target resolutions. For our application at higher resolutions, and particularly with an interest for strongly heterogeneous precipitation patterns at the ridge-scale, a less-diffusive advection scheme was required. The issue of numerical diffusivity in complex terrain has been well documented (Westerhuis et al., 2021; Lundquist et al., 2012). Higher order advection stencils (odd-ordered up to 5th order) have thus been implemented in the HICAR model. These schemes, in combination with the SLEVE coordinate system Schaer2002,Kruyt2022, reduce numerical diffusion in HICAR simulations. To achieve larger physics time steps, a pseudo-Runge-Kutta-3 (RK3) advection integration is added to HICAR (Wicker and Skamarock, 2002). Lastly, the use of RK3 time stepping required the addition of a monotonic flux-limiter for the standard advection scheme (Wang et al., 2009).

Since the original publication of G16, numerous physics parameterizations have been added to the model, and will be detailed in Kruyt et al., 2023, *in prep.*. Of importance to this paper, the Noah land surface model (LSM) (Ek et al., 2003), Morrison microphysics scheme (Morrison et al., 2009), RRTMG radiation scheme (Thompson et al., 2016), and the YSU PBL scheme (Hu et al., 2013) have all been added to the model and will be used for the simulations which follow in later sections.

### 2.4 Asynchronous I/O

As model efficiency increases, it is natural to push the model to run for larger domains and larger time periods. Additionally, as the simulation resolution increases, forcing data of a higher resolution is needed. The cumulative effect of these two points is that efficient, high-resolution models must output and input large amounts of data (Prein et al., 2015). For example, for the setup used in section 4.2.1, one day of simulation requires reading 11GB of forcing data and outputting 14.5GB of data, depending on output variables selected. To avoid blocking I/O operations on the runtime loop and to facilitate a many-programs one-file access pattern, an asynchronous I/O strategy was adopted. This is shown in Figure 1 via the blue elements on the left. Input and output is handled by a few processes which are split from the simulation processes at initialization. These I/O processes then coordinate their file access through parallel netCDF I/O, resulting in less demand on the file system and eliminating the need for stitching together output files in post-processing. These changes make the model faster by overlapping I/O with physics processes, and make it possible to directly use simulation output to force one-way nested runs, as done in section 4.2.1.

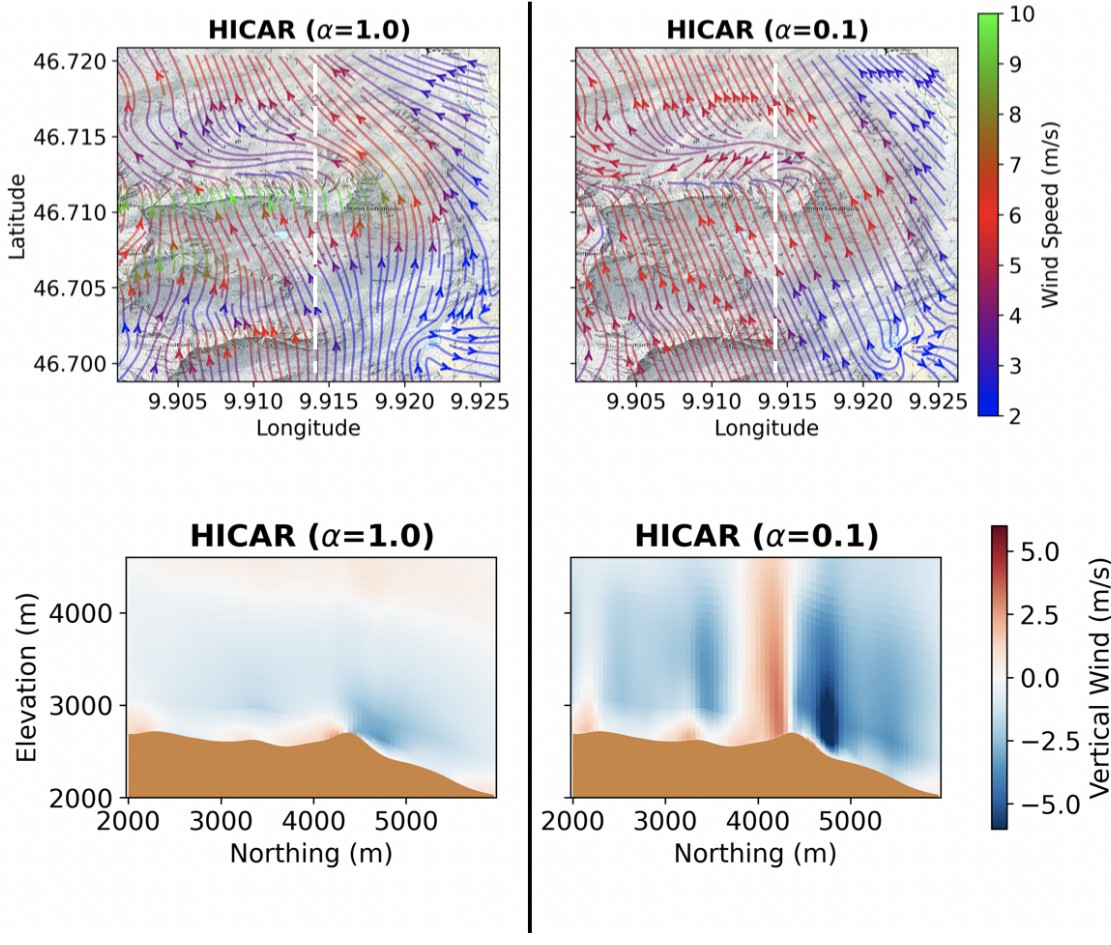

**Figure 4.** Demonstration of the two end-member solutions for HICAR's wind solver under the two extreme stability conditions. The plan view panels in the top row are centered on a ridge cutting horizontally across the figure. A vertical transect across this ridge is shown in the lower panels, with the location of the transect indicated in the upper panels by the white dotted line. Surface wind flow lines are overlaid on a topographic base map in the upper panels, with flow line color corresponding to wind speed. The left column of the figure displays the maximum stable condition, while the right column shows the maximum unstable condition.

## 3 Model Setup and Datasets

### 3.1 COSMO Model

The COSMO model is run operationally by the Swiss weather service, MeteoSwiss, over a domain encompassing Switzerland
(www.cosmo-model.org). COSMO is a non-hydrostatic, compressible atmospheric model capable of simulating the state of the atmosphere over complex terrain such as the Swiss Alps. Predicted variables from COSMO such as temperature, humidity, and wind speeds are made available by MeteoSwiss. Output from the 1.1km and 2km resolution COSMO simulations, COSMO1





and COSMO2, respectively, are used in this study. COSMO2 output is used to force the 1350m WRF, ICAR, and HICAR simulations discussed in section 4.1 and 4.2.1, while COSMO1 output is used to force the 250m HICAR simulation in section 4.2.2 and 4.3, and the 450m HICAR simulation in section 4.4. The HICAR simulations are forced with specific humidity, temperature, pressure, and the 3-D wind field (U/V/W) from the COSMO model. All COSMO variables are bi-linearly interpolated in 3D to the HICAR grid using latitude, longitude, and vertical height. Then, specific humidity and temperature are forced at the boundaries, while pressure and winds are input for the full 3-D grid, with the winds being further modified using the downscaling scheme described in section 2.

## 3.2 WRF Model

The Weather Research and Forecasting (WRF) model (Skamarock et al., 2008) is a non-hydrostatic and compressible atmospheric model used widely in research and operational forecasting (Benjamin et al., 2016). WRF has also been successfully run at very high resolutions (50m) over the complex terrain of the alps (Gerber et al., 2018, 2019; Goger et al., 2022; Kruyt et al., 2022). For these reason, we use WRF in this study to demonstrate a "gold-standard" for atmospheric modeling in comparison to HICAR runs. All output from the WRF model comes from prior simulations first presented in Gerber et al. (2018), and thus guided the choice of spatio-temporal domain for some of the simulations presented in section 4.

## 3.3 ICAR/HICAR Setup

Simulations using the ICAR and HICAR models, introduced in section 2, are presented in section 4. The HICAR simulations utilize the YSU PBL scheme, the Noah land surface model, RRTMG radiation scheme, and the Morrison two-moment microphysics scheme. This microphysics scheme was chosen due to its demonstrated efficacy in forecasting precipitation in complex terrain (Liu et al., 2011), and use in the WRF simulations of Gerber et al. (2018). Only the wind fields from the ICAR simulations are analyzed, and because there is no physics-dynamics coupling in either ICAR or HICAR, ICAR was not run with these physics parameterizations enabled.

HICAR has been developed as a variant of the ICAR model, as these models share a core code base. The HICAR variant of ICAR can be turned on by passing "HICAR" to the variant option of the namelist file. This switches on a number of namelist options, ensuring that the configuration is optimized for high-resolution runs in complex terrains. Specifically, the namelist options which designate a run with the HICAR model include: terrain-following SLEVE coordinates, variational-calculus-based wind solver, and wind modifications based on terrain-descriptors.

## 3.4 Gridded Datasets

In section 4.2.2, two gridded datasets for precipitation are used, MeteoSwiss's RhiresD product (MeteoCH, 2013), and the precipitation product produced by the SLF Operational Snow Hydrology Service (OSHD) using an Optimal Interpolation (OI) technique (Magnusson et al. (2014); Mott et al., submitted). RhiresD is constructed by taking precipitation data from a dense network of precipitation gauges distributed throughout the Alps, and then applying a climatological precipitation-elevation





gradient to extrapolate observations beyond gauges, using a version of the PRISM algorithm (Daly et al., 1994). The OSHD

product is obtained by first partitioning RhiresD into solid and liquid precipitation and then updating the snowfall fraction by assimilating snow station data from 350 locations using optimal interpolation (Magnusson et al., 2014). This allows for a higher station density at higher elevations relative to RhiresD, and minimizes underestimates of precipitation during snowfall events due to gauge undercatch. Of course, selecting for snow station sites introduces other spatial biases in station representativeness (Grünewald and Lehning, 2015). A full description of the OI procedure used in the OSHD product can be found in Mott et al.,

submitted.

## 3.5 Spatio-temporal Domains

Sections 4.1 and 4.2.1, as well as the figures presented in section 2, use the same 50m domain introduced in Gerber et al. (2018). It is roughly 10km x 10km square, with the 50m horizontal resolution simulations covering a 24 hour period over the day of March 5th, 2016. This domain covers the Upper Dischma valley outside of Davos, Switzerland. We adopt the terminology "xx

315 m simulation" to refer to the horizontal resolution of a simulation. The 50m HICAR and ICAR simulations for this run are nested within 150m, 450m, and 1350m simulations of the same respective model, following the methodology of Gerber et al. (2018) for their WRF runs. Importantly, ICAR/HICAR allows the use of a coarser vertical grid than WRF (Horak et al., 2021). As a result, the WRF simulations use 40, 40, 60, and 90 vertical levels for the 1350m, 450m, 150m, and 50m simulations, while ICAR/HICAR used only 20, 20, 60, and 60.

Sections 4.2.2 and 4.3 discuss results from a 250m simulation of HICAR covering most of the Swiss Alps from Lausanne in the west to Val Müstair in the east, for a roughly 280km X 170km domain. The simulation was run for the month of January 2017.

Section 4.4 repeats a benchmarking setup from Kruyt et al. (2022), running the HICAR model at a 50m resolution for five days in March 2019 over a roughly 7.5km x 7.5km domain. This 50m domain is nested within a 450m domain, following the

325 methodology of Kruyt et al. (2022).

High-resolution domain data for all simulations comes from Gerber and Lehning (2021), which provides ASTER Global Digital Elevation Model V002 and Corine land use data at a resolution of 1 arcsec (Spacesystems and Team, 2019; Agency, 2006). For the HICAR simulations, this terrain data was then upscaled to the desired target resolution with no smoothing applied. In order to run the WRF model at resolutions approaching 50m, certain considerations must be applied to the model

topography. For the WRF simulations, to ensure model stability at reasonably long time steps, the terrain for all high-resolution simulations is smoothed using a 1-2-1 smoothing filter with 14 passes, and the terrain near the boundaries of the outer-most domain is smoothed to match the COSMO topography. Although this smoothing procedure is not required to run ICAR/HICAR, the same smoothed terrain data as the WRF simulation is used for one HICAR simulation presented in section 4.2.1. This is done in order to enable a direct comparison between WRF and HICAR for the same topography.





## 4  Model Demonstrations

### 4.1  Wind Fields

In section 2.2, the effects of the changes to the wind solver were shown for comparison with ICAR (Figure 3) and for a demonstration of their ability to simulate atmospheric stability (Figure 4). To discuss the wind solver of HICAR in the context of existing atmospheric models, we present here results comparing HICAR to the WRF model. Figure 5 shows a plan view of multiple model simulations at 50m over complex terrain in the Upper Dischma valley of Davos, Switzerland. As discussed in section 2, the COSMO forcing data provided is expected to capture the effects of mountain waves which the linear wind solver of ICAR is designed to capture, so this module of ICAR was turned off. As a result, the ICAR simulation shown is bilinearly interpolated COSMO2 data. The surface flow field from ICAR is quite homogenous as a result, with uniform south-westerly flow over the domain and a narrow range of wind speeds over the domain. This is in contrast to the WRF simulation, which reports various modifications to the flow pattern (blocking, cross-slope flow, terrain-induced speed-up), as well as a larger range of wind speeds. This result is instructive that ICAR alone is not suitable for high-resolution simulations. WRF also reports higher wind speeds at ridge crests than any of the HICAR simulations, but WRF has been found to overestimate speed up of winds over topography (Gerber et al., 2018; Gómez-Navarro et al., 2015).

For examining the effects of the wind solver detailed above, we present two HICAR simulations: one with the empirical adjustments based on terrain-descriptors and one without. The simulation without terrain-descriptors uses a procedure to diagnose its winds which is similar to that employed by models like WindNinja (Forthofer et al., 2014) but, with the distinction of using a spatio-temporally varying value for $\alpha$ (EQ #17). This simulation already captures a wider range of surface wind speeds than the base ICAR model, and offers some of the flow field deflection observed with the WRF model. This is consistent with prior studies which have employed the technique from Sherman (1978). Once the terrain descriptors are used, we see that certain features of the flow field present in the WRF simulation also emerge in the full HICAR run. Of note are the cross-slope flows and lee-side reductions in wind speed. Due to the improved terrain representation capable with the ICAR/HICAR model, these flow features develop for secondary valleys not fully resolved in the WRF topography. This demonstrates the added value of this two-step approach to generating a diagnostic, mass-conserving wind field.

The advantages of the terrain descriptors are on show in Figure 6 as well. This figure presents a vertical cross section of modeled flow across the Sattelhorn ridge, which is in the upper-center of Figure 5. The WRF model shows a large eddy in the lee-side of the ridge, with a long horizontal extent and reduced wind speeds relative to the flow outside of the lee. This eddy also gives rise to up-slope flow at the surface of the lee of the ridge. The HICAR run simulates a similar dynamic structure. The eddy present in HICAR has a shorter horizontal extent and is stronger, resulting in higher wind speeds within the eddy and faster up-slope flow at the surface of the lee. Despite these differences in the properties of the eddy, the ability of HICAR to predict the presence of such flow features is a surprising result, since no prior applications of Sherman 1978's technique have reported such behavior. We attribute this to our use of terrain-descriptors, which predispose the solution of Sherman 1978 to generate an eddy in the lee, all of which may be due to the sharper terrain represented by HICAR. It is easy to imagine how this approach of pre-conditioning a wind field and then using a diagnostic, mass-conserving solver, could be used to parameterize



other dynamic effects, and has previously been shown to yield reasonable results when parameterizing thermally driven winds
(Forthofer, 2007). We also note that the calculation of the terrain-descriptor based corrections depends upon somewhat arbitrary
constants, and thus could be adjusted to yield eddies of varying horizontal extent. This tuning of the terrain-descriptor-based
adjustments will be done in a future study, using distributed observations of winds in complex terrain as a basis for tuning and
validation.

The differences in terrain representation between WRF and ICAR/HICAR are also on display in Figures 5 and 6. WRF and
375 other models which prognostically solve for winds rely on spatial gradients of pressure to calculate wind speeds. In order to
simplify the lower boundary condition, these models also typically employ terrain following coordinates where model coor-
dinate surfaces slope as the terrain does. This means that high-resolution simulations will feature large coordinate distortion,
and pressure differences in the horizontal may become quite large as one vertical cell surface exists at lower elevations than
another. This may lead to large pressure gradients which require very fine time steps to stably integrate. The model terrain is
380 typically smoothed to allow for smaller grid distortions, smaller pressure gradients, and thus larger time steps. Recent imple-
mentation of an immersed boundary method in WRF allows for this entire consideration to be skipped, although such a domain
discretization comes with its own trade-offs (Lundquist et al., 2012).

The above discussion is valid for atmospheric models which solve prognostic equations for momentum. Neither the ICAR
model nor the HICAR variant do this, opting for diagnostic solutions for the wind field instead. As a result, issues of model
stability arising from terrain steepness do not exist, and we can include model terrain without any artificial smoothing or
implicit numerical diffusion. This is apparent in the elevation profile of Figure 6 and, to a lesser extent, in the DEM of Figure 5.
The difference in terrain used may lead to the different lee-side dynamics when comparing the HICAR and WRF simulations.
This ability of ICAR and HICAR to represent the terrain without any artificial smoothing is a major strength of both models.
High-resolution atmospheric modeling is assumed to yield more accurate forecasts in part through improved representation of
390 the underlying terrain. If HICAR can represent topography more accurately than WRF at the same horizontal resolution, and
without implicit numerical diffusion, it gives the model an "effective" resolution larger than that of WRF.

## 4.2 Precipitation Distribution

### 4.2.1 Ridge-scale

The above discussion of terrain representation also plays an important role in precipitation distribution, as is on display in
Figure 7. There are noticeable differences in the snowfall transects of the two HICAR simulations, one using the unsmoothed
topography (HICAR) and the other using WRF's smoothed topography (HICAR, WRF-topo). This result supports the above
point that HICAR's improved terrain representation leads to a higher effective model resolution, impacting the simulation
results. We also note a strong wet-bias over the domain for the WRF model, with precipitation amounts nearly double what
was recorded at a snow depth station located in the domain (Figure 7). The snowfall transects reveal ridge-scale differences in
precipitation for all model simulations, with the windward (left) side of the ridge receiving approximately 15% more snowfall
than the leeward (right) side in the HICAR simulations. The WRF simulation shows a similar although more modest ridge-scale



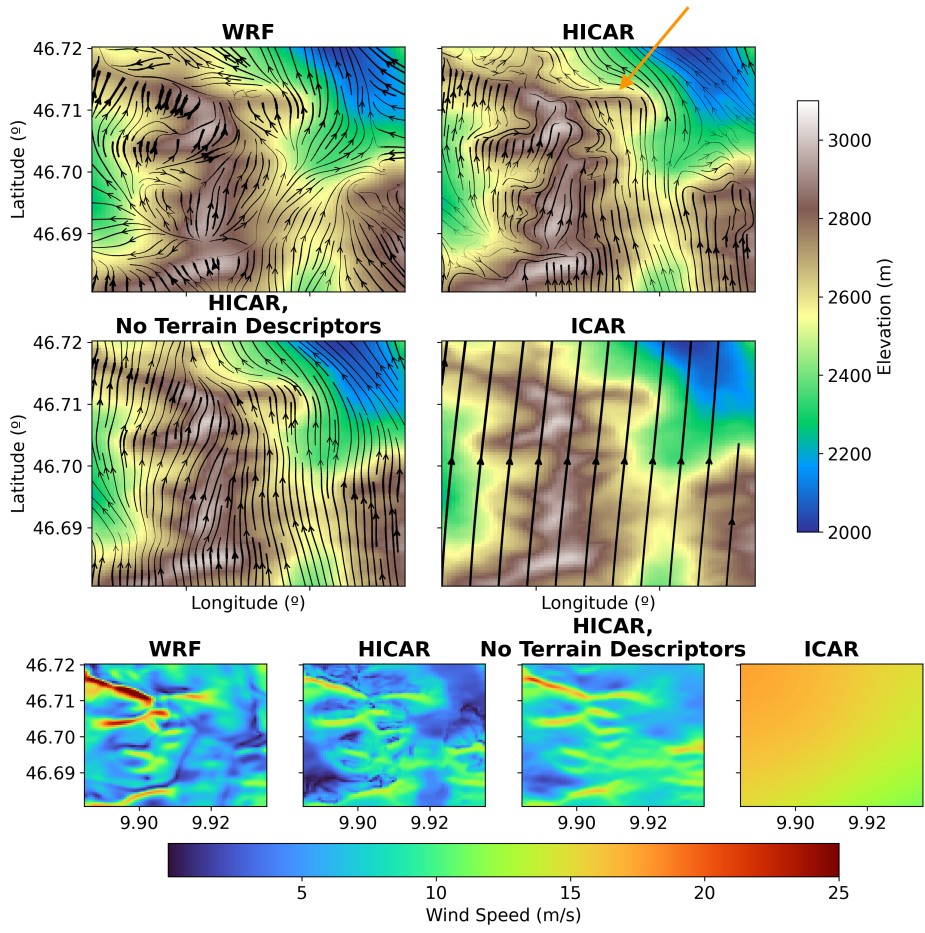

**Figure 5.** Comparison of surface flow fields at a 50m resolution between models and model setups for March 5th, 2016, 00:00 UTC+1. The upper four panels show flow fields overlaid on model topography. Model topography is smoothed for the WRF run compared to the HICAR/ICAR runs. Thickness of flow lines corresponds to wind speed, with thicker flow lines indicating higher wind speeds. The lower row of panels displays the surface wind speeds of the various model runs. The sparser flow lines for the ICAR simulation are a plotting decision to avoid redundancy and do not reflect a difference in the simulation setup. The orange arrow indicates the location of the Sattelhorn Ridge, which is shown in profile in Figure 6.

difference, with a positive snowfall anomaly (relative to mean over the transect) beginning on the windward side and continuing until just downwind of the ridge, followed by a steady decrease in snowfall anomaly. The main difference between the HICAR and WRF simulations are the magnitude of the windward and leeward differences. This can be partly explained by the leeside





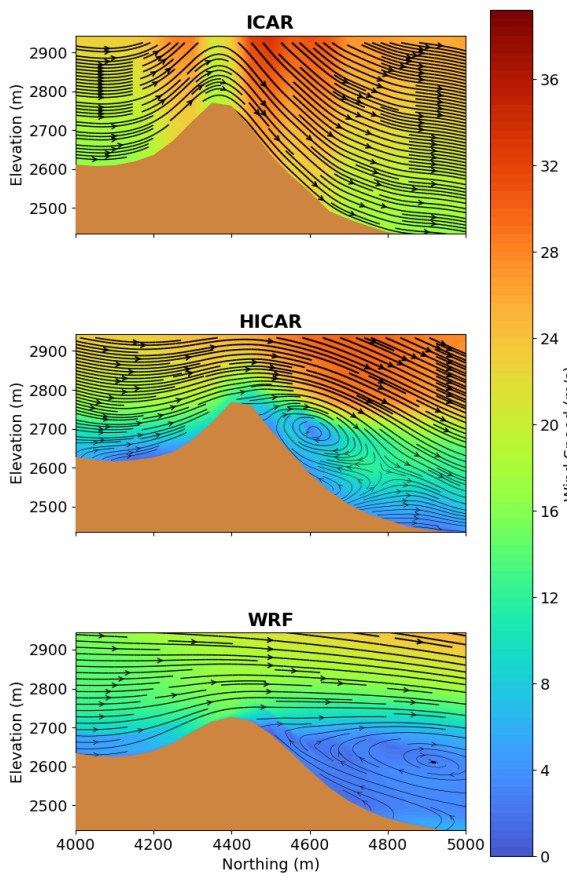

**Figure 6.** Profile view of flow fields at a 50m resolution between models for March 5th, 2016, 02:00 UTC+1. Wind direction is indicated by the flow lines, and line thickness corresponds to wind speed, where thicker lines show higher wind speeds. Wind speed is given by the background color. A profile of the underlying terrain is shown in each panel, with the WRF simulation having smoother terrain than the ICAR or HICAR simulations.

dynamics simulated by both models. Taking the flow profiles shown in Figure 6 to be representative of the flow differences over the 24-hour event, we note that HICAR has higher wind speeds aloft on the leeside of the ridge due to the presence of the eddy. The peak in precipitation on the windward side is likely due to blocking of the lowlevel flow and reduced wind speeds on this side of the peak (Figure 6). We note a positive anomaly in snow depth just downwind of the ridge, which we attribute to the strong horizontal wind speeds aloft, inline with previous studies of preferential deposition (Mott et al., 2014; Wang and



Huang, 2017). In fact, the HICAR snow depth distributions show a similar windward/leeward pattern to results obtained by Comola et al. (2019) using an LES model over ideal topography. This cumulative effect of the flow field on snow depth can be realized intuitively by tracing the flow lines of Figure 6 across the ridge and imagining snow sedimentation given a constant sedimentation rate. The question of if this flow pattern is accurate for this particular event has not been demonstrated, but given the proven accuracy of HICARs advection scheme (Wang et al., 2009), the resultant deposition pattern is certainly physically

consistent with the given flow field. This discussion demonstrates the research utility of HICAR: it can be used to efficiently (Section 4.4) test different flow patterns at the ridge scale and see how they affect particle-flow interactions. A later validation of HICARs flow fields would determine how predictive the simulated deposition patterns are.

### 4.2.2    Range-Scale

Accurate high-resolution precipitation estimates in complex terrain are a slippery target (Lundquist et al., 2019; Bonekamp

et al., 2018). Gauge-based gridded products are subject to gauge undercatch, and assumptions about the spatial patterns used to interpolate them (Rasmussen et al., 2012; Collados-Lara et al., 2018; Lundquist et al., 2010). Radar products meanwhile suffer from occlusion when scanning in complex terrain (Germann et al., 2022). As a result, high-resolution comparisons of modeled versus observed precipitation in complex terrain deserve careful consideration to offer any form of model validation. We spare any detailed quantitative validation for a future study, and instead offer a comparison of different gridded precipitation products

for the sake of discussion.

   Figure 8 shows accumulated precipitation for January 2017 from two gridded products and a 250m HICAR simulation. We first note that the majority of storms during January 2017 came from the northwest, and our simulation domain for HICAR extended slightly beyond the boundaries of the figure shown to just include the Swiss Plateau. The HICAR simulation is forced with only water vapor from COSMO1, so the microphysics requires some time to "spin-up", generating hydrometeors and thus

precipitation. This may explain some of the lower precipitation amounts along the pre-Alps in the upper northwest of the figure relative to both RhiresD and the OSHD precipitation product.

   Overall, Figure 8 shows remarkable agreement between HICAR and the two gridded precipitation products for a one month winter period. The OSHD precipitation product gives larger precipitation values at higher elevations than RhiresD since it is generated by back-calculating precipitation from snow water equivalent, avoiding gauge undercatch during snowfall events

(Magnusson et al., 2014). This result suggests that the larger precipitation values obtained from the HICAR simulation are possible. The inter-alpine areas (center) of the domain however show less precipitation in HICAR than either gridded product, especially in the valleys. However, these differences between HICAR and the other gridded products are comparable to differences observed between the gridded products themselves. Lastly, we note that the product using climatological averages for its interpolation, RhiresD, returns a smoother field of precipitation than either HICAR or OSHD OI. The OI product yields

stronger elevation gradients of precipitation, which is likely due to its higher station density at higher elevations relative to RhiresD, and its ability to capture unbiased precipitation during snowfall events. This suggests that the stronger gradients observed from HICAR are appropriate. None of this discussion is to assert an accuracy of one product over another, but is instead



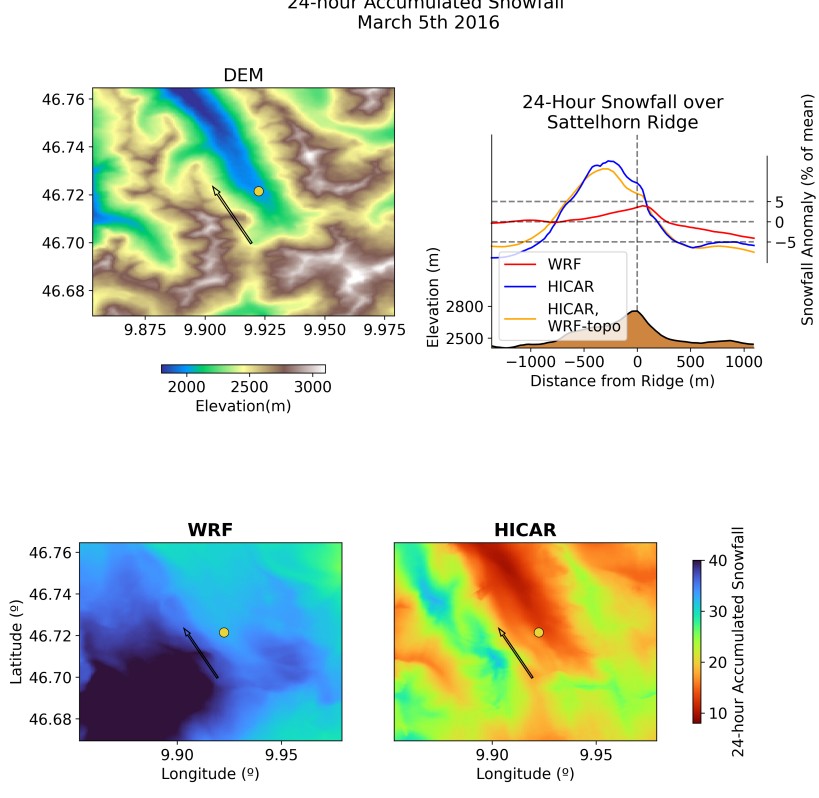

**Figure 7.** Differences in snowfall over the Upper Dischma Valley for a 24-hour snowfall event on March 5th, 2016. All terrain data displayed is from the unsmoothed HICAR run. All values of snowfall are reported in cm, with the WRF and HICAR snowfall values converted from mass to depth assuming a constant density of 100 kg m-3. The upper left panel shows a DEM of the area, with a dot in the valley indicating the location of a snow depth sensor an arrow indicating the location and direction (left-right) of the transect shown in the upper right panel. This arrow points along the prevailing wind direction during the 24-hour snowfall. The upper right panel shows snow depth transects across the Sattelhorn ridge for three model simulations, WRF, HICAR, and HICAR run with the same smoothed topography as WRF. Mean snowfall is almost twice as large in WRF than in HICAR, so snow depth is reported as percentage of the mean snow depth along the transect in order to compare the HICAR and WRF simulations on the same graph. The lower two panels show the spatial distribution of snow depth across the domain, with the value recorded at the snow depth station over the 24-hour period (20.3cm) overlaid.

to demonstrate that HICAR's precipitation estimate is as consistent with existing precipitation products as those products are with each other.

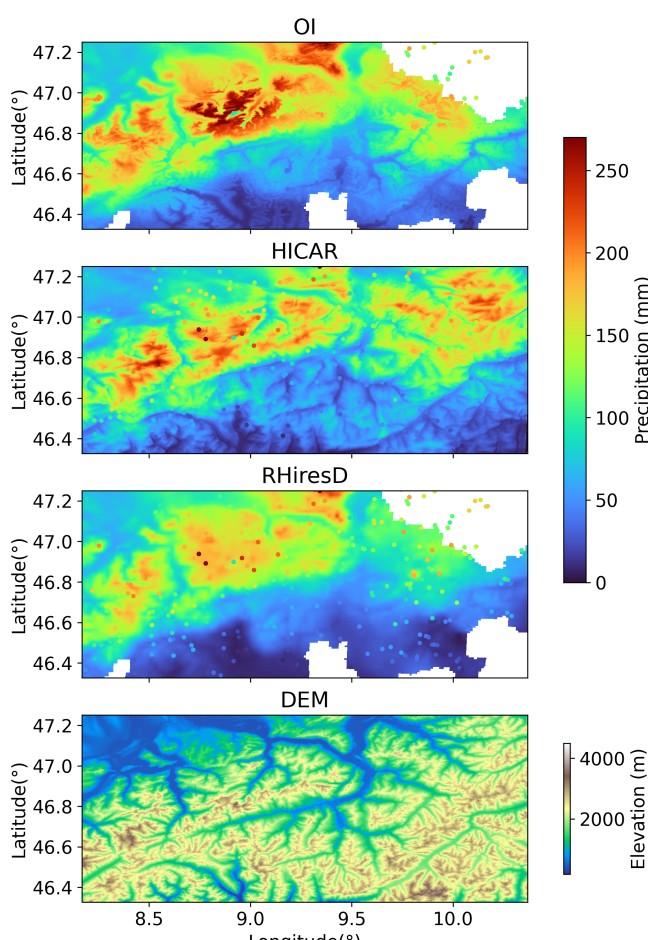

**Figure 8.** Precipitation over the central and eastern Swiss Alps during January 2017 at a 250m resolution. All three plots of precipitation have point data from the OI product overlaid as dots. Since these mostly coincide with the same values for the OI product, the dots are often indistinguishable from the background field in the top panel.

## 4.3 Cold Air Pooling

Figure 9 shows a cold air pooling event on the morning of January 24th, 2017. We observe that, over the course of the early morning hours, strong mesoscale winds recede from over the valley, allowing a cool, stable surface layer to develop and for that cool air to migrate toward lower elevations. This surface layer is ultimately re-mixed as wind speeds increase and surface





**Table 1.** Core-hours per simulation day for benchmarking run

|                    | WRF    | ICAR  | HICAR |
|--------------------|--------|-------|-------|
| Core-hours         | 33,993 | 1,336 | 288   |
| Speed-up over WRF  | 1.0    | 25.4  | 118.0 |

cooling decreases around 9 AM local time. These results are somewhat surprising, as a parameterization of thermally driven
flows is not yet included in HICAR. Thus, the flow patterns shown are largely unaware of the evolving thermal stratification of
the valley. However, the wind solver used in HICAR is designed to minimize differences between its wind field and the wind
field supplied from the forcing data. If the driving model, in this case COSMO1, simulates valley winds supportive of cold air
pooling, and the LSM of HICAR simulates a cooling of the surface, cold air pooling as shown in Figure 9 is possible. This
figure demonstrates an important part of the HICAR model: its dependency on physically consistent winds from forcing data.
Future parameterizations of thermally driven winds may make the model independently capable of simulating phenomena such
as cold air pooling.

### 4.4 Computational Efficiency

The main reason why HICAR may be attractive as a model is through its computational efficiency relative to existing atmo-
spheric models such as WRF or COSMO. Aside from HICARs improved representation of terrain, the model is not expected
to simulate physical phenomena better than more complex models. Thus, understanding its computational demand is central
to establishing its utility. To quantify this demand, we repeat a benchmarking setup described in Kruyt et al. (2022). We run
HICAR at a 50m resolution over a roughly 7.5 x 7.5 km domain for a 5 day period in March 2019, which includes several
winter storms. The model numerics/physics setup is the same as those used for the above subsections for which results are
shown. The results of the benchmarking test are presented in Table 1, alongside the results previously published in Kruyt et al.
(2022). The main takeaway from this comparison is that HICAR uses 118x fewer computational resources than WRF for the
same simulation. Stated otherwise, a year of simulation over this domain with WRF would require a significant allotment of
computing time ( 350,000 node hours, assuming 36 cores per node). With HICAR, the same simulation represents a fraction
of a modest project allocation ( 3,000 node hours).

The more than four-fold speedup of HICAR relative to ICAR is also somewhat surprising. This result is best explained by
the switch from the GNU fortran compiler to the Cray compiler. Testing of Coarray fortran, on which ICAR is parallelized
(Rasmussen et al., 2018), has revealed the Cray compiler to have a faster implementation of this fortran standard than GNU.
Since 70-80% of HICAR's runtime is consumed by message passing, this results in a considerable speed up between the two
compiler types. Additionally, the high-performance computing architecture used in this study is a Cray computer. The use of a
native compiler may contribute to speed up as well.

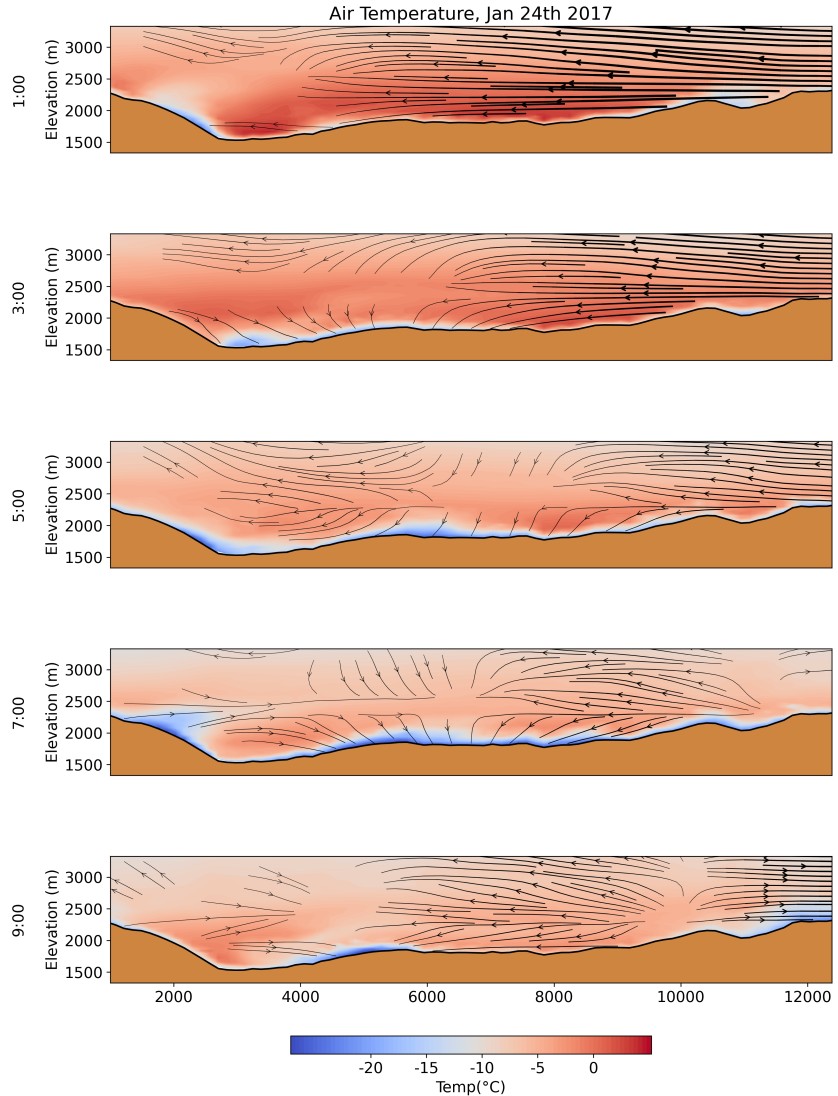

**Figure 9.** The development and diffusion of surface cooling for an alpine valley during dawn. The plot shows a small area of the 250m Swiss Alp domain introduced in section 3.5. The local time is indicated on the y-axis label. Wind vectors are plotted for wind directions along the transect. Thicker vectors indicate higher wind speeds, and winds below 0.2 m s-1 are not plotted.

## 475  5  Conclusions

In this paper we have introduced the High-resolution variant of the ICAR model, HICAR. We detailed its primary modifications to adapt it for simulations over high-resolution complex terrain. This consists primarily of a new approach to solving for a 3D wind field which utilizes terrain-descriptors, TPI and Sx, to pre-condition the input wind field to approximate some expected



effects of the topography on the flow field (Figure 2). These effects are parameterized simply and rely on assumptions and
somewhat arbitrary constants. The model's sensitivity to these constants will be further investigated in a future study. After
this correction step, the pre-conditioned wind field is fed into an optimization routine, which makes the resulting field mass-
conserving while minimizing changes to the pre-conditioned field (Figure 1). A novel approach to the diagnostic wind solver
is adopted which allows atmospheric stability to influence the solution as it varies in time as well as space. This allows for
low-level flow blocking, leeside recirculation, and cross-slope flows to be simulated by the model. These changes to the wind
solver, in addition to a new advection scheme and physics parameterizations, enable the results demonstrated in section 4.

We observe a marked improvement in the representation of wind fields in complex terrain over the base ICAR model when
comparing against the WRF atmospheric model (Figure 5). These improvements to the wind field make HICAR capable of
simulating heterogenous snow deposition patterns in complex terrain, which show clear signals resulting from terrain-flow
interactions (Figures 6 and 7). At larger scales, precipitation patterns in complex terrain are represented to the same goodness
as existing gridded precipitation products (Figure 9). ICAR/HICAR also forgoes any consideration of pressure gradients in its
dynamics, allowing it to be run without any smoothing of the underlying terrain. Most importantly, all of these developments
were done while maintaining the orders of magnitude speed up over WRF which ICAR originally demonstrated. The result
is a model which is 118x faster than WRF and can run at very high resolutions (50m), extending intermediate complexity
atmospheric modeling into the resolutions typically used by land surface modelers. HICAR's ability to handle very steep
terrain, coupled with its computational speed, seems well suited for modeling efforts over High Mountain Asia, where testing
of various model configurations is already performed with more computationally expensive models (Bonekamp et al., 2018).
HICAR's computational efficiency also enables high resolution simulations over long time scales, supporting climate impact
studies at the regional scale and seasonal studies of coupled glacier-atmosphere or snow-atmosphere models at hectometer
scales. This last point will be expanded upon in future publications, where HICAR will be coupled with an intermediate
complexity snow model to enable high-resolution forecasting of winter snowpack and spring melt. This will involve the addition
of a thermal wind parameterization to improve surface flows over glaciers and snow (Mott et al., 2020), with the goal of better
resolving advective surface-atmosphere processes such as turbulent heat exchange. As atmospheric models begin to regularly
probe higher resolutions, HICAR enables rapid testing and iteration of various model configurations with relatively little
computational cost. This makes HICAR a powerful companion to conventional atmospheric models.

*Code and data availability.* HICAR can be used for non-profit purposes under the GPLv3 license (http://www.gnu.org/licenses/gpl-3.0.html,
last access: 1 February 2023). Code for the model is available at https://github.com/d-reynolds/HICAR (INSERT GITHUB PAGE FOR ICAR
ONCE HICAR CHANGES MERGED). The exact release (v1.0) used in this publication is available at https://doi.org/10.5281/zenodo.7610241.
The model has dependencies for the netCDF4-parallel fortran and PETSc libraries. Paralellisation is achieved through fortran Coarrays,
which utilizes different message passing protocols depending on the compiler used. For use with the GNU fortran compiler, OpenCoarrays
is required.



*Author contributions.* DR implemented the model changes detailed in the paper. EG developed the base ICAR model and provided regular feedback on development. BK developed input and analysis scripts used to run the model and assisted with troubleshooting the initial application of ICAR to high resolutions. MH provided feedback when developing near-surface wind fields. FG provided the WRF runs used in the study and assisted in designing the experimental setup of HICAR runs. ML and RM gave regular direction regarding the scope and 515 aims of the study. DR wrote the manuscript with contributions from the other authors.

*Competing interests.* The contact author declares that none of the authors have any competing interests.

*Acknowledgements.* The authors thank the funding source of this project, the Swiss National Science Foundation grant #188554. The computational resources needed to perform the simulations were provided by the Swiss National Supercomputing Center (CSCS) through projects s1148 and s999. The authors would like to thank Jean-Marie Bettems and Petra Baumann for their helpful correspondence when working 520 with COSMO data hosted by MeteoSwiss. Developers of open source python toolboxes, particularly xarray and xesmf, have also played a crucial role in this study by enabling efficient analysis and manipulation of large datasets.





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
