# Peer review of "The High-resolution Intermediate Complexity Atmospheric Research (HICAR v1.1) Model Enables Fast Dynamic Downscaling to the Hectometer Scale"

_Geoscientific Model Development, 2023_

## Referee Comment (RC1)

**The High-resolution Intermediate Complexity Atmospheric Research (HICAR v1.0) Model Enables Fast Dynamic Downscaling to the Hectometer Scale**

**Referee Comments**

This work introduces a new high-resolution version of the Intermediate Complexity Atmospheric Research (ICAR) model. At grid spacings on the hectometric range, terrain effects on the wind field are inevitable, therefore new terrain descriptors are implemented in the model to modify the near-surface wind speed. As input data, either COSMO-1 or COSMO-2 runs are used, allowing the authors to avoid downscaling based on linear wave theory and the divergence problem mentioned by Horak et al. (2021). HICAR exhibits a good performance compared to WRF simulations, which are used as a "reference": HICAR is able to show lee-side eddies, realistic precipitation patterns and realistic cold-air pool development. Some of these capabilities of HICAR were surprising for the authors; the major advantage of HICAR is, however, the light use of computational resources. The manuscript is well-written and gives a comprehensive overview over the changes made to suit ICAR for high resolutions. However, improvements in the presentation of the results are necessary, namely replacing the current colormaps in the Figures. Furthermore, some discussion points should be made, especially the dependency on high-resolution input data. When these changes are implemented in the manuscript, it can be accepted for Geoscientific Model Development.

**General comments**

1. Introduction.The authors point out the great advantage of HICAR in terms of saving computational resources. However, the authors might also mention that ICAR does not require topography smoothing as opposed to standard NWP models. The more realistic topography has a positive impact on simulated wind speeds.

2. Figure quality (color schemes). Please avoid using the "rainbow", "jet", "brg" or other miscellaneous color maps in Figs 4-8 (exception: topography plots). These colormaps are outdated outdated, because they are not perceptually uniform

and do not give useful information about variable quantities (Stauffer et al. 2014). For example, in Fig 7, the colormap is completely contradictory to the results shown. Please change to a sequential colormap and completely avoid the aforementioned colormaps. If you use matplotlib in Python, you can easily choose a sequential colormap from here: `https://matplotlib.org/stable/tutorials/colors/colormaps.html`

3. The authors are surprised by the good performance of HICAR, which is perhaps every model developer's dream. However, as the authors mention in the manuscript as well, HICAR seems to profit stronlgy from the high-resoultion input data from either COSMO-1 or COSMO-2. I wonder whether these positive effects on lee-side eddies and cold-air pools would be also present in ICAR with a 'coarser' input dataset, for example from IFS, ERA-5, or a coarser-resolution WRF run. Did the authors perform test runs with coarser input data? This might be of interest for mountainous regions where high-resolution NWP model output is sparse (i.e., the Himalayas or the Andes). After reading the manuscript, HICAR only seems to work with an already high-resolution input data set ($\Delta x < 3$ km), because linear wave theory can be avoided. Please discuss this necessity of a high-resolution input data set in the manuscript in more detail.

4. The manuscript would profit from an overview table on the different model setups of ICAR, with which one has smoothed topography or not, and which setup useses physics coupling, etc. The table would make it easier to remember the contents of Section 3.5 while reading about the results.

**Specific comments**

- line 249: Put the references in a bracket

- line 284: these reasonS

- line 348: The overestimation of wind speeds by WRF has been also reported by Umek et al. (2021) and Goger et al. (2022).

- line 356,367: I think this is one of the major advantages of HICAR. You can highlight this in both the introduction and the conclusions.

- line 363: Up-slope flow. In mountain meteorology, the term "up-slope flow" is usually attributed to thermally-induced flows due to differential heating. I would re-word it to "reverse flow", since the lee-side eddy seems to be dynamically-induced.

- line 391: What do you mean exactly by effective resolution?

- line 447, "surface layer": Do you mean stable boundary layer?

- line 453: COSMO is actually successfull in simulating thermally-induced flows associated with cold air pools (Goger et al. 2018, their Fig. 3). Another method to "check" whether a NWP model is able to simulate the revelant ABL processes in a valley is to check whether at least ten grid points are present along a cross-section between the ridges (Wagner et al. 2014).

- line 474: You can mention Piz Daint and its specifications within this paragraph if you wish.

- line 505: I think you can remove the text in capslock

**References**

Goger, B., M. W. Rotach, A. Gohm, O. Fuhrer, I. Stiperski, and A. A. M. Holtslag, 2018: The impact of three-dimensional effects on the simulation of turbulence kinetic energy in a major alpine valley. *Boundary-Layer Meteorol*, **168 (1)**, 1–27, doi:10.1007/s10546-018-0341-y.

Goger, B., I. Stiperski, L. Nicholson, and T. Sauter, 2022: Large-eddy simulations of the atmospheric boundary layer over an alpine glacier: Impact of synoptic flow direction and governing processes. *Q. J. R. Meteorol. Soc*, **148 (744)**, 1319–1343, doi:https://doi.org/10.1002/qj.4263.

Horak, J., M. Hofer, E. Gutmann, A. Gohm, and M. W. Rotach, 2021: A process-based evaluation of the intermediate complexity atmospheric research model (icar) 1.0.1. **14 (3)**, 1657–1680, doi:10.5194/gmd-14-1657-2021.

Stauffer, R., G. J. Mayr, M. Dabernig, and A. Zeileis, 2014: Somewhere Over the Rainbow: How to Make Effective Use of Colors in Meteorological Visualizations. *Bull. Amer. Meteor. Soc.*, **96 (2)**, 203–216, doi:10.1175/BAMS-D-13-00155.1.

Umek, L., A. Gohm, M. Haid, H. C. Ward, and M. W. Rotach, 2021: Large eddy simulation of foehn-cold pool interactions in the Inn Valley during PIANO IOP2. *Q. J. R. Meteor. Soc.*, **147 (735)**, 944–982, doi:10.1002/qj.3954.

Wagner, J. S., A. Gohm, and M. W. Rotach, 2014: The Impact of Horizontal Model Grid Resolution on the Boundary Layer Structure over an Idealized Valley. *Mon. Wea. Rev.*, **142 (9)**, 3446–3465, doi:10.1175/MWR-D-14-00002.1.

---

## Author Comment (AC1)

We thank both anonymous reviewers for their feedback regarding this manuscript. The comments have helped to clarify the discussion, focus our conclusions, and has enlightened us to the use of the newest, perceptually uniform colormaps. Find below a response to the major comments from the two reviewers. Responses to the minor comments are included where necessary. If not directly responded to, they have been accepted and incorporated into the text.

**Reviewer #1**

1. Introduction. The authors point out the great advantage of HICAR in terms of saving computational resources. However, the authors might also mention that ICAR does not require topography smoothing as opposed to standard NWP models. The more realistic topography has a positive impact on simulated wind speeds.

An additional sentence has been added to the end of section 3.4 to make this point explicitly, as well as the second sentence of the conclusion section.

2. Figure quality (color schemes). Please avoid using the "rainbow", "jet", "brg" or other miscellaneous color maps in Figs 4-8 (exception: topography plots). These colormaps are outdated outdated, because they are not perceptually uniform and do not give useful information about variable quantities (Stauffer et al. 2014). For example, in Fig 7, the colormap is completely contradictory to the results shown. Please change to a sequential colormap and completely avoid the aforementioned colormaps. If you use matplotlib in Python, you can easily choose a sequential colormap from here: https://matplotlib.org/stable/tutorials/colors/colormaps.h

Thanks for this note – we had originally used the turbo color scheme (https://ai.googleblog.com/2019/08/turbo-improved-rainbow-colormap-for.html), believing that it was perceptually uniform, but upon closer reading it is not. To remedy this, we have changed all of the former offending colormaps to one from Fabio Crameri's preceptually uniform colormap library (https://www.fabiocrameri.ch/batlow/).

3. The authors are surprised by the good performance of HICAR, which is perhaps every model developer's dream. However, as the authors mention in the manuscript as well, HICAR seems to profit stronlgy from the high-resoultion input data from either COSMO-1 or COSMO-2. I wonder whether these positive effects on lee-side eddies and cold-air

pools would be also present in ICAR with a 'coarser' input dataset, for example from IFS, ERA-5, or a coarser-resolution WRF run. Did the authors perform test runs with coarser input data? This might be of interest for mountainous regions where high-resolution NWP model output is sparse (i.e., the Himalayas or the Andes). After reading the manuscript, HICAR only seems to work with an already high-resolution in-put data set (x < 3 km), because linear wave theory can be avoided. Please discuss this necessity of a high-resolution input data set in the manuscript in more detail.

Additional discussion has been added to the end of section 4.3. This is certainly a necessary take-away for the reader. We have not tried running with forcing data besides COSMO-1 or COSMO-2. We do have a version of the model with the linear mountain wave solver and the variational wind solver detailed here, where linear wave perturbations are applied to the wind field before passing the wind field through the variational solver. As noted though, the authors believe that a proper examination of the resolution of forcing data at which HICAR is no longer useful would add too much to this model introduction paper.

4. The manuscript would profit from an overview table on the different model setups of ICAR, with which one has smoothed topography or not, and which setup useses physics coupling, etc. The table would make it easier to remember the contents of Section 3.5 while reading about the results.

Added (Table 1)

L391: Reworded to avoid the term "effective resolution" as laid out by Skamarock 2004. Arguing that the model has a larger effective resolution would require an energy spectrum analysis, which is not done here. Instead, we want to highlight that a 50m run with lots of smoothing applied to the topography is more likely to give results similar to a 250m run than a 50m run, since the topography has been smoothed to filter out high-frequency features.

**Reviewer #2**

Section 2.3: To be honest, I don't fully understand how authors get the x118 speedup relative to the WRF model. HICAR employs RK3 integration to solve advection in addition to the physics

parameterizations. But, Mielikainen et al. (2014) in GMDD (https://doi.org/10.5194/gmdd-7-8941-2014) showed that dynamic core takes 61.93% of the computation time of WRF. If I take into account the cumulus scheme in addition to dynamics, I expect the integration speed to increase roughly by a factor of 2 at maximum. Is it because of the difference in the compiler?

The speedup reported is possible due to a number of reasons. First, the dynamics of HICAR are massively simplified in comparison to WRF. The only "dynamics" considered by HICAR are the solving of the wind field, which is only computed once at each input time step, in this case once every 1hour of simulation time. The remaining dynamics-related process is the advection module. This reduces model runtime itself by removing additional processes to solve for (i.e. pressure solver). Additionally, the wind field of HICAR produces lower wind speeds than the WRF model, allowing for a larger model time step via the CFL criterion. The improved numerical stability of HICAR also allows for fewer vertical levels near the surface of the terrain, cutting down on model elements to solve for. Lastly, as now mentioned explicitly in the text under section 4.4, we have extensively tested various syntaxes and message passing strategies with Fortran Coarrays, finding a best-possible configuration which allowed for a 3x increase in speedup alone compared to the earlier draft of this manuscript. Gutmann et al. (2016) also found a speedup of 100-800x compared to WRF.

The model changes discussed in this paper do not appear to degrade performance of HICAR compared to ICAR for a number of reasons. 1) The wind solver introduced consumes a negligible percentage of the overall computation time. This is because terrain-descriptors are computed at initialization. 2) The RK3 advection scheme does demand calling the advection code 3-times per time step, but also allows increasing the timestep by up to 1.6 times compared to non-RK3 advection. This increase in time step allows for more infrequent calls to the physics parameterizations, which as mentioned above, are a significant portion of the runtime. Lastly, 3) I/O has effectively disappeared from model runtime through the use of asynchronous I/O. Thus, it is reasonable that the model with such changes alone should be comparable to the speed-ups reported by Gutmann et al. (2016). Additional optimization steps to the core runtime-code and to the message-passing method further solidify this point.

Line 252- 255: What is the rationale for using the YSU PBL scheme? Is it recommended to turn off the PBL scheme at the hectometer scale? Also, what do you use for the surface-layer scheme?

The YSU PBL scheme is used due to it's prior use in studies of atmospheric modeling in the gray-zone, especially during winter months. Yes, it is often recommended to turn off the PBL scheme at some scale-length, or to use a scale-aware scheme, but these considerations follow from the notion that the dynamics of the model have resolved some amount of turbulent mixing. The wind fields generated by ICAR/HICAR are temporally fairly invariant. One wind field is solved for at the current input step, and a second is solved for at the next input step. For the physics steps in between, the wind field is interpolated through time between the two wind fields at the two bookending input time steps. This means that the idea of resolved-scale turbulence is unlikely to apply or behave in a similar fashion to models like WRF. So existing best-practices regarding PBL schemes in the gray-zone are not likely to transfer when running HICAR. This question of which PBL scheme to apply, if at all, is a topic which we are interested in investigating further in the future.

A comment about surface layer scheme used in ICAR/HICAR now added in section 3.3.

Line 398-399: Why do you see wet bias over the entire domain in the WRF model? Is it coming from parent domains?

The wet bias in WRF observed over the domain is believed to be a function of excessively strong orographically enhanced precipitation for this domain, as discussed in Gerber et al. (2018). A sentence mentioning this has been added to section 4.2.1.

Section 4.2: How much are snow and precipitation different between ICAR and HICAR? It would be very nice to have a discussion on this.

The difference in precipitation, and thus snow as well, between ICAR and HICAR is frankly so different at these model resolutions that the results are not particularly interesting. ICAR's wind field and advection scheme result in a fairly diffuse field of moisture and temperature, with strong vertical updrafts (Figure 3). These serve to produce either little precipitation over the domain, or excessive fallout of precipitation on the windward edge of the domain. The results shown in Kruyt et al. (2022) were obtained by coupling the hydrometeor fields from the COSMO1 model to ICAR, effectively bypassing the formation of hydrometeors in ICAR itself. For the setup shown here, where a series of nested domains are used and the outermost nest is forced with only winds, temperature, moisture, and pressure, the ICAR model did not generate fields of precipitation reasonable for comparison. This was perhaps the guiding reason for the development of HICAR.

Eq 1: What is the difference from the Laplacian operator?

We are not sure which equation is referred to here – EQ#1 does not resemble the Laplacian operator.

Line 269: Please add the full name and citation for the COSMO model.

The COSMO model is often cited in this way (Winstral et al. (2017)) – no singular "COSMO" citation appears to exist.

Line 470: Did you compare HICAR to WRF compiled with the Cray compiler? The same compiler should be used to be a fair comparison.

We agree that a proper comparison of the WRF and HICAR models would involve a test with both models compiled on the same compiler. However, for this study we did not have sufficient computing resources to re-do the benchmarking runs mentioned in section 4.4, and thus simply report the core-hour usage reported in Kruyt et al. (2022). It is worth noting that Kruyt et al. (2022), as well as Gutmann et al. (2016), did not report the compilers used when comparing the WRF and ICAR models, although Kruyt et al. (2022) used the Intel compiler for WRF and the GNU compiler for ICAR. A discussion of this point has been added to section 4.4

Section 5: I find it would be nice if the authors discuss the possibility to use their model in the wind energy field. This is not a demand, merely a wish.

Thanks for the comment – we agree that this would be an interesting discussion point, especially around the ability of the model to represent the vertical structure of the boundary-layer. However, to keep the scope of the paper focused, we have left out this discus